# Approach to the Adult Acute Lymphoblastic Leukemia Patient

**DOI:** 10.3390/jcm8081175

**Published:** 2019-08-06

**Authors:** Valentina Sas, Vlad Moisoiu, Patric Teodorescu, Sebastian Tranca, Laura Pop, Sabina Iluta, Sergiu Pasca, Cristina Blag, Sorin Man, Andrei Roman, Catalin Constantinescu, Ioana Rus, Mihail Buse, Bogdan Fetica, Mirela Marian, Cristina Selicean, Ioana Berindan-Neagoe, Bobe Petrushev, Horia Bumbea, Alina Tanase, Mihnea Zdrenghea, Shigeo Fuji, Shigehisa Kitano, Ciprian Tomuleasa

**Affiliations:** 1Department of Hematology, Iuliu Hatieganu University of Medicine and Pharmacy, Cluj-Napoca 400124, Romania; 2Department of Pediatrics, Iuliu Hatieganu University of Medicine and Pharmacy, Cluj-Napoca 400124, Romania; 3Department of Pediatrics, University Emergency Pediatrics Hospital, Cluj Napoca 400124, Romania; 4Department of Hematology, Ion Chiricuta Clinical Cancer Center, Cluj-Napoca 400124, Romania; 5Intensive Care Unit, University Emergency Hospital, Cluj Napoca 400124, Romania; 6Research Center for Functional Genomics and Translational Medicine, Iuliu Hatieganu University of Medicine and Pharmacy, Cluj-Napoca 400124, Romania; 7Department of Radiology, Ion Chiricuta Clinical Cancer Center, Cluj-Napoca 400124, Romania; 8Department of Radiology, Iuliu Hatieganu University of Medicine and Pharmacy, Cluj-Napoca 400124, Romania; 9MedFuture Center for Advanced Medicine, Iuliu Hatieganu University of Medicine and Pharmacy, Cluj Napoca 400124, Romania; 10Department of Pathology, Ion Chiricuta Clinical Cancer Center, Cluj-Napoca 400015, Romania; 11Department of Pathology, Regina Maria Diagnostic Laboratory, Cluj Napoca 40008, Romania; 12Department of Pathology, Octavian Fodor Regional Institute of Gastroenterology and Hematology, Cluj Napoca 400111, Romania; 13Department of Hematology, Carol Davila University of Medicine and Pharmacy, Bucharest 020021, Romania; 14Department of Stem Cell Transplantation, Fundeni Clinical Institute, Bucharest 020021, Romania; 15Department of Hematology and Stem Cell Transplantation, Osaka International Cancer Institute, Osaka 5378511, Japan; 16Department of Experimental Therapeutics, National Cancer Center Hospital, Tokyo 1040045, Japan; 17Division of Cancer Immunotherapy, Exploratory Oncology Research and Clinical Trial Center, National Cancer Center, Tokyo 1040045, Japan

**Keywords:** acute lymphoblastic leukemia, clinical management, follow-up

## Abstract

During recent decades, understanding of the molecular mechanisms of acute lymphoblastic leukemia (ALL) has improved considerably, resulting in better risk stratification of patients and increased survival rates. Age, white blood cell count (WBC), and specific genetic abnormalities are the most important factors that define risk groups for ALL. State-of-the-art diagnosis of ALL requires cytological and cytogenetical analyses, as well as flow cytometry and high-throughput sequencing assays. An important aspect in the diagnostic characterization of patients with ALL is the identification of the Philadelphia (Ph) chromosome, which warrants the addition of tyrosine kinase inhibitors (TKI) to the chemotherapy backbone. Data that support the benefit of hematopoietic stem cell transplantation (HSCT) in high risk patient subsets or in late relapse patients are still questioned and have yet to be determined conclusive. This article presents the newly published data in ALL workup and treatment, putting it into perspective for the attending physician in hematology and oncology.

## 1. Background on ALL Work-Up and Follow-Up

Acute leukemias are classified into acute myeloid (AML) and acute lymphoblastic leukemias (ALL), each form having a characteristic immunophenotype. Based on the cytological aspect of the blasts, there are three main types of acute lymphoblastic leukemia: L1, with small cells that have a large nucleus; L2, with larger, pleomorphic blasts; and L3, with a highly basophilic cytoplasm. The cytological classification has now been mostly replaced by the World Health Organization (WHO) classification, which divides ALLs into B-cell ALL and T-cell ALL. B-cell ALL and acute lymphoblastic lymphomas are malignancies with B-cell lymphoblasts [1,2,3]. When the primary disease is diagnosed in a lymph node, the correct name is “acute lymphoblastic lymphoma.” In B-cell ALLs (B-ALLs), which represent around 85% of all pediatric ALLs, the bone marrow aspirate displays at least 25% bone marrow lymphoblasts [1,2,3,4]. Despite tremendous improvements in understanding the molecular mechanisms behind B-cell ALL (B-ALL), the prognosis of these patients is rather poor, especially for old or frail patients that are unable to withstand aggressive chemotherapy or allogeneic hematopoietic stem cell transplantation. Moreover, in the case of adult patients, the disease is often already disseminated in extramedullary sites, especially in the central nervous system. Still, for young adults, the prognosis has been significantly improved, as proven by the Group of Research on Adult ALL (GRAALL) randomized controlled trial, which explored the role of hyperfractionated cyclophosphamide (hyper-C) dose intensification in newly-diagnosed Philadelphia (Ph)-negative ALL patients on a chemotherapy regimen similar to that of pediatric patients [5]. The complete remission (CR) was 91.9%, and for a median follow-up of 5.2 years, the 5-year event-free survival (EFS) and overall survival (OS) were 52.2% and 58.5%, respectively. In adolescents and young adults aged 15–20 years, the use of full pediatric protocols is supported by many comparative studies, with long term-survival of almost 70% [6].

The initial diagnosis of a possible malignancy is determined based on clinical symptoms such as weight loss, night sweats, fatigue, infections, and bleeding, as well as by altered abnormal laboratory results that indicate anemia or thrombocytopenia. Subsequently, the final diagnosis is based on flow cytometry immunophenotyping that identifies malignant clones positive for cluster of differentiation (CD) 10, CD19, CD20, CD22, CD24, and CD79a. During normal B cell maturation, CD34 is first downregulated together with terminal deoxynucleotidyl transferase (TdT), followed by CD10 and CD38, while CD45 expression is upregulated along with CD21 and CD22 (11244048). Most aberrancies are related to the co-expression/over- or under-expression of CD10, TdT, CD38, CD34, CD20, and cross lineage myeloid expression, while aberrant T-cell antigen expression is less frequent.

Patients with chromosomal alterations such as hyperploidy or t(12;21)(p13;q22) have a better prognosis, whereas t(9;22)(q34;q11.2) or t(4;11)(q21;q23) translocations are associated with poorer survival rates [7,8,9,10]. In addition, the 2016 WHO classification includes two provisional indicators associated with negative prognosis: B-ALL with intrachromosomal amplification of chromosome 21, and BCR-ABL1-like B-ALL. The latter is defined by BCR-ABL1 ALL-like alteration of the IKZF1 gene without the BCR-ABL1 fusion protein. The genomic landscape of BCR-ABL1-like B-ALL suggests that this entity is characterized by alterations in a limited number of pathways, all of which are responsive to inhibition with existing TKIs [10,11,12,13]. 

Conversely, T-cell ALL is less frequent, with immunophenotype analysis that identifies cells positive for CD1, CD2, CD4, CD5, CD8 and CD10. In pediatric patients, the prognosis of T-cell ALL is excellent, exemplified by the 90–95% of cases that achieve complete remission (CR) after chemotherapy. Of great importance for the diagnosis and prognosis of ALL is determination of the presence of the *BCR-ABL* transcript [10]. Philadelphia-positive (Ph+) patients, who historically were high risk cases, now benefit from TKIs, with significantly improved prognoses. 

Complete remission is generally defined by: fewer than 5% blasts in the bone marrow; normal maturation of all cellular components in the bone marrow; no extramedullary disease (e.g., CNS, soft tissue disease); ANC (absolute neutrophil count) of at least 1000/µL; platelets more than 100,000/µL; and, lastly, transfusion-independent patients [14]. The detection of residual cells by flow cytometry following therapy is based on asynchronous expression of antigens in comparison with a normal maturation pattern [15], as further discussed later in the manuscript.

## 2. Standard Chemotherapy Regimens 

Chemotherapy for ALL is administered in accordance to Ph+ status [16]. First-line chemotherapy for patients under 65 years diagnosed with ALL is the Hoeltzer protocol, which consists of seven steps described in Figure 1 and Table 1 [17,18]. A valid alternative for therapy is the HyperCVAD protocol, which consists of two treatment cycles repeated four times and an additional maintenance chemotherapy cycle (refer to Figure 2 and Table 2) [19,20,21,22,23,24].

For relapsed ALL patients under 65, the salvage chemotherapy regimen is based on three blocks of administration with methotrexate (MTX), as shown in Figure 3 and Table 3. For T-cell ALLs that relapse after the first cycle of chemotherapy, the protocol includes the administration of 1.5 mg/m^2^ nelarabine at Days 1, 3, and 5, a protocol that is repeated six times. Salvage chemotherapy for these patients also includes a high dose of cytarabine (Cyt) in combination with mitoxantrone protocol that consists of the administration of 3 mg/m^2^ Cyt at Days 1–5 plus 80 mg/m^2^ mitoxantrone at Day 3 [25,26]. Ph+ ALLs are also treated with a protocol that consists of an induction cycle, two consolidation cycles, and one maintenance cycle, as shown in Figure 4 and Table 4 [27]. This maintenance chemotherapy cycle consists of 600 mg imatinib each day for the remainder of the regimen, as further discussed in the manuscript.

An important variable in ALL outcome is age. While adolescents and young adults (from 15/18 to 35/40 years old) are treated with higher intensity and higher cumulative doses of drugs, elderly patients require a less aggressive protocol based on much lower doses of corticosteroids, vincristine and asparaginase, with the avoidance of anthracyclines and alkylating agents to reduce treatment related mortality (TRM). ALL affects many patients worldwide and requires long and rigorous courses of chemotherapy in three stages, with the maintenance phase lasting 2–3 years. While the primary drugs used in the maintenance phase, 6-mercaptopurine (6-MP) and methotrexate (MTX), are required to decrease the risk of relapse, they also have potentially serious toxicities, including myelosuppression, which may be life-threatening, as well as gastrointestinal toxicity [28]. Modern therapies, including immunotherapy, have shown good results in the treatment of ALL. Still, cytokine release syndrome (CRS) and neurotoxicity are common toxicities, potentially life-threatening in severe cases. Risk factors for CRS and neurotoxicity identified so far include disease burden, lymphodepletion intensity, and the chimeric antigen receptor T cells (CAR-T) cell dose administered. Risk-adapted dosing, with lower doses administered to patients with high marrow blast counts, has been successful at decreasing severe CRS rates in this population. Drugs such as tocilizumab and corticosteroids have been effective at ameliorating toxicity, enabling CAR-T cells to be administered safely to many patients without significantly compromising efficacy. A deeper understanding of the pathophysiology underlying CRS and neurotoxicity will enable the development of novel approaches to reduce toxicity and improve outcomes [29,30,31,32,33], as discussed further.

For older and/or frail patients, therapy consists of 1 g/kg prednisone for 30 days along with 600–800 mg/day of imatinib, with the latter therapy continued long-term. For patients over 65 years, induction chemotherapy includes the administration of 2 mg vincristine intravenous (i.v.) at Days 1, 8, 15, and 28, plus 25 mg/m^2^ adriblastine i.v. at Days 1, 8, 15, and 28, plus 8 mg/m^2^ dexamethasone or 1 mg/kg prednisone at Days 1–28, plus intrathecal administration of 15 mg methotrexate at Day 1. Maintenance is achieved with purinethol or MTX administration for two years.

In B-ALL, CR is defined as the absence of any symptoms related to the dissemination of blasts into the lymph node or internal organs, and peripheral blood counts of more than 1500 neutrophils/mmc, more than 100,000 platelets, and more than 10 mg/dL hemoglobin. The blood smear must display less than 5% blasts, with neither cytological nor cytogenetic abnormalities. Hematopoietic stem cell transplantation (HSCT) is designated to B-ALL patients in CR after first-line chemotherapy has failed, in line with the ESMO Clinical Practice Guidelines [34]. Patients may undergo HSCT as first-line treatment after the first cycle of chemotherapy if in CR and exhibiting negative prognostic factors. Such prognostic factors include more than 40,000 leukocytes at diagnosis or high-risk genetic abnormalities [35]. Recent protocols use measurable residual disease (MRD) < 0.01% to define MRD as positive. MRD assessment and cytogenetics are important prognostic factors in all patients treated upfront with chemotherapy, routinely used to risk-stratify cases and make decisions regarding whether to move on to an allogeneic stem cell transplantation (SCT) or immunotherapy [36,37,38,39], as seen in Figure 5.

## 3. Management of CNS Involvement

Another important aspect for ALL patients is the involvement of the central nervous system (CNS). Data in support of prophylaxis are limited and most physicians use additional measures [40,41]. Thus, lumbar puncture may be carried out, the cerebrospinal fluid investigated, and multiple doses of intrathecal chemotherapy with high-dose MTX and/or Cyt administered. Whether high-dose chemotherapy alone is enough to properly prevent CNS prophylaxis, or whether it should be associated with radiotherapy, is a subject of debate. Physicians should consider both the anti-leukemia effects of CNS relapse prophylaxis and the long-term effect on the brain tissue. In Poland, Zając-Spychała et al. reported that ALL patients treated with high-dose chemotherapy according to the ALL IC-BFM 2002 trial had cognitive impairment and a decreased volume of selected subcortical structures [42]. Another option is a very high dose of MTX (33.6 g/m^2^), which resulted in similar OS to other CNS-directed therapies without the long-term impact on cognitive functions, but with substantial acute toxicities [43].

CNS prophylaxis can be achieved by CNS irradiation, intrathecal MTX in mono- or triple therapy (MTX, Cyt and steroids), and systemic high-dose therapy with MTX and/or cytarabine. CNS involvement at diagnosis requires both a standard chemotherapy model and intrathecal chemotherapy until spinal fluid cytology shows blast clearance. Intrathecal therapy consists of MTX 12.5 mg, Cyt 50 mg, and prednisone 40 mg on Days 1 and 15 of Courses 1, 2, and 8 and on Day 1 of Courses 4, 6, and maintenance months 2, 3, 4, and 5 (×12) [44]. When using very high-dose MTX, Nathan et al. reported that dose adjustments were needed in 74 of the 88 patient cohort [43]. Out of the 74 patients with dose adjustments, 9 cases had impaired MTX clearance or renal disfunction, 5 had hepatic toxicities, 1 patient had seizures, and 1 had pulmonary toxicity. None of the patients died because of the therapy. Still, this therapy is less toxic than cranial radiation plus intrathecal MTX; the patients treated with very high-dose MTX had a stable verbal IQ and increase in performance IQ related to expected practice events over time, in comparison with the group treated with radiotherapy plus intrathecal MTX [45,46].

Recurrences within the CNS usually coincide with or predict systemic relapse in the marrow and blood [47,48]. Even if adults achieve CR in most cases, relapses are diagnosed in 7% of cases, especially in the elderly and high-risk populations [49,50]. Without prophylactic intrathecal chemotherapy, more than half of adult ALL patients have CNS involvement or relapse [51]. Dara et al. reported that in patients treated with intrathecal chemotherapy following first-line intrathecal chemotherapy with methotrexate (MTX) and corticosteroids, CR was achieved in 76% of patients. Moreover, 91% reached CR after second-line intrathecal chemotherapy [52]. Clinical response was documented in 75% of cases. Although most patients were additionally treated with systemic chemotherapy, response rates did not differ between patients treated with CNS-penetrating and non-CNS-penetrating drugs. CNS progression/relapse occurred in 40% of patients, with a median progression-free survival (PFS) of 12.2 months. The median OS was 18.3 months and 55% of the patients died during follow-up. CNS relapse remains a major problem for the ALL patient [53], one that will probably be better managed in the near future with the introduction of novel therapies, such as blinatumomab or CAR-T cells [54,55,56,57].

Various risk factors have been associated with CNS involvement in ALL, whether at initial diagnosis or at relapse. CNS involvement has a higher incidence in younger patients. Cerebrospinal fluid (CSF) examination is the most important laboratory examination, and pathologists associate relapse after initial clearing with increased opening pressure, elevated protein (>50 mg/dL), decreased glucose (<60 mg/dL), and an increased WBC count (>5/mm^3^). These altered parameters are also common in infectious disease, such as in bacterial or viral meningitis. The presence of leukemic cells in the CSF is diagnostic for CNS relapse, a scenario defined as unequivocal morphological evidence of leukemic blasts in the CNS and/or mononuclear cell count > 5/µL. In children, HSCT did not appear to improve outcomes. Thus, Eapen et al. showed that when comparing 149 patients enrolled on clinical trials to 60 HLA-matched sibling transplanted patients, all treated over a 10-year period, TRM rates were higher following transplantation [58]. The 8-year probabilities of leukemia-free survival were similar after chemotherapy with irradiation and transplantation (66 and 58%, respectively). In the absence of an advantage for one treatment option over another, the data support use of either intensive chemotherapy with irradiation or HLA-matched sibling transplantation with total body irradiation containing a conditioning regimen for children with ALL in second remission after an isolated CNS relapse. 

## 4. Tyrosine Kinase Inhibitors for Ph-Positive ALL

TKIs are used in combination with classic chemotherapy for Ph-positive ALL [59,60,61]. At the University of Texas MD Anderson Cancer Center (MDACC), Jabbour and Kantarjian et al. showed that the combination chemotherapy of hyper-CVAD with ponatinib is efficient in achieving early sustained remissions in Ph-positive ALL [62]. The 2-year event-free survival rate was 81%. Grade 3 or more toxic effects included infections during induction, reported in 54% of patients, as well as increased hepatocytolysis in 38% of patients, thrombotic events in 8% of cases, myocardial infarction in 8% of cases, hypertension in 16% of cases, skin rash in 22% of cases, and pancreatitis in 16% of patients. Continuing the investigation, the same MDACC group in Houston showed that when comparing hyper-CVAD plus ponatinib with hyper-CVAD plus dasatinib, the clinical outcome of patients that had received hyper-CVAD plus ponatinib had superior results [63]. The 3-year event-free survival (EFS) for patients treated with hyperCVAD plus ponatinib and hyperCVAD plus dasatinib were 69% and 46%, respectively (*p* = 0.04), and the 3-year OS rates were 83% and 56%, respectively (*p* = 0.03). 

The availability of imatinib and of second generation TKIs has improved the outcome of elderly Ph+ ALL patients. When treated with imatinib, such cases have a better OS in comparison with Ph-negative ALL [64,65]. Advanced age and comorbidities are associated with high TRM. Treatment with a reduced-intensity induction and imatinib yields CR rates with no induction mortality, but with a very high rate of relapse if consolidation is not given [64], resulting in poor OS. With second generation TKIs, lower intensity chemotherapy after induction brings forward similar outcomes as with intensive consolidation, but still poor long-term survival [66,67]. 

In a randomized trial that compared standard versus less-intensive chemotherapy for newly diagnosed Ph+ ALL patients, Chalandon et al. showed that less intensive treatment reduced the early mortality without affecting the efficacy, with higher CR rates and an equivalent major molecular response rate (MMR) [68]. MMR is defined as a BCR-ABL1/ABL ratio less than 0.1% in the BM. They analyzed 270 newly diagnosed Ph+ ALL patients aged 18 to 59. Following the enrolment, patients were randomized to receive imatinib combined with either the less intensive chemotherapy based on vincristine and steroids or the more intense regime with cyclophosphamide, vincristine, doxorubicin, and steroids. In the second course, all patients received similar treatment with MTX, Cyt, and imatinib, followed by HSCT when suitable. The primary endpoint was the assessment of MMR after the second course, which was 66.11% for the less intensive treatment and 64.5% for the more aggressive treatment. CR was higher in the first arm (less intensive) in comparison with the second one (98.5% versus 91%). EFS, relapse-free survival, cumulative incidence of relapse, and OS did not differ between arms. The GRAAAL experience of combining imatinib with standard chemotherapy was afterwards confirmed by other subsequent investigators [69,70,71], with both German and Italian investigators looking at whether intensive chemotherapy during induction might not be as efficient as initially thought, a dilemma still in debate. In the transplant setting, the administration of a TKI does not influence the NRM, but is correlated with an increased frequency of acute graft versus host disease (GVHD) [72,73]. Thus, imatinib might be very efficient in the treatment of chronic GVHD, especially when sclerotic and fibrotic features are described. Using TKIs has significantly improved the clinical outcome of Ph+ ALL for the patients that undergo a HSCT, a scenario most likely to improve once novel immunotherapy options are introduced, as is the case of the T-cell-engaging bispecific antibody blinatumomab [74].

For Ph-negative ALL, the role of imatinib has yet to be investigated properly in large clinical trials. Still, a link between imatinib-based therapy and the development of a Ph-negative ALL was reported by Cherrier-De Wilde et al. [75]. The describe a case of a chronic myeloid leukemia (CML) treated at the first line with imatinib that subsequently developed a Ph-negative ALL, suggesting that the leukemic blasts were Ph-negative while residual Ph-positive cells were detected by PCR. Thus, the case described was rather a secondary leukemia and not a blast crisis of CML. Still, the 2019 data do not provide enough proof that might support adding imatinib to Ph-negative ALL.

For R/R T-cell ALL patients, who do not benefit from blinatumomab, alternative salvage therapy consists of nelarabine, an FDA-approved, T-cell-specific purine nucleoside analogue. Nelarabine is also currently being studied along with γ-secretase inhibitors blocking Notch1 signaling in T-cell ALL. In a phase 2, open-label, multicentric trial, nelarabine was given in an alternate-day cycle (Days 1, 3, and 5) at 1.5 g/m^2^/day for R/R T-cell ALL, with a new cycle every 22 days. The results showed a rate of CR rate of 31% (95% CI, 17–48%) and 1-year OS of 28%. The introduction of TKIs into Ph+ ALL treatment has improved CR rates from 60–70% to 80–90% simply by adding imatinib to standard chemotherapy regimens. The optimal usage of TKIs is still being debated: which TKI-based chemotherapy combination is the best, whether using second-generation TKIs (nilotinib, dasatinib) should be preferred over imatinib, or if the standard intensive chemotherapy/TKI combination has any advantage over reduced-dose chemotherapy with TKI.

Ponatinib is a multitargeted TKI [76] with action against Ph-positive lymphoid malignancies [77], especially the cases with *BCR-ABL* T315I mutation [78,79,80]. Ph+ ALL was associated with cure rates of less than 25% before the TKI era, but present-day therapy leads to long-term survivals for more than 75% of cases [81]. Ponatinib is very efficient, as proven by the pivotal phase 2 ponatinib Ph+ ALL and CML Evaluation (PACE) trial, which evaluated its efficacy at a starting dose of 45 mg once daily in patients resistant/intolerant to dasatinib or nilotinib due to their harboring the *BCR-ABL1* T315I mutation, with durable and meaningful clinical responses in the heavily pretreated patients [82].

Recent studies have raised several issues regarding allogeneic hematopoietic stem cell transplantation (HSCT): Do HSCT benefits outweigh procedure-related acute mortality and long-term effects in the case of adolescents and young adults? Should minimal residual disease (MRD) be used to determine HSCT or to continue chemotherapy after remission? How should we identify the appropriate type of HSCT and associated conditioning regimens (matched unrelated donor (MUDs), sibling donor, umbilical cord, haploidentical, reduced intensity chemotherapy (RIC)-HSCT)? 

## 5. Allogeneic Stem Cell Transplantation for ALL

HSCT should be part of the consolidation therapy for patients with unfavorable molecular profiles. HSCT is part of the consolidation strategy for the ALL patient, as post-remission consolidation strategies include chemotherapy and HSCT [83]. Thus, the management of such cases depends on the patient and disease characteristics, especially in the era of novel targeted therapies, such as blinatumomab, inotuzumab, ozogamycin, and CAR-T cells [54,55,64]. Novel drugs may provide a bridge to transplant for the ALL patient [84,85] and HSCT is curative for such cases. Nevertheless, the benefit of consolidation chemotherapy remains uncertain in these cases [86]. Thus, when investigating 261 adults with relapsed/refractory (R/R) ALL examined across two phase 2 studies that received blinatumomab in cycles of 4 week continuous infusion and 2 week treatment-free intervals, 56% achieved CR during the first 2 cycles, compared with 46% of younger adults. 

HSCT in the case of standard-risk adults has a less clearly defined role with the advent of MRD as a prognostic marker capable of easily restratifying patients to high-risk. While in patients with molecularly undetectable leukemia, no proven survival benefit conferred by allo-HSCT was shown when compared to standard chemotherapy, for positive MRD, allogeneic HSCT was linked with an improved relapse-free survival [87,88].

Even if HSCT is curative for an ALL patient in CR1, the benefits of consolidation chemotherapy do not appear to add any significant benefit for patients with available donors who undergo a myeloablative HSCT, as proven by the study coordinated by Weisforf et al. [89]. In 524 adult patients with ALL in CR1 who received at least two, one, or no cycles of consolidation before myeloablative HSCT from 2008 to 2012, those receiving at least two, one, or no cycles of consolidation had an adjusted 3-year cumulative incidence of relapse of 20%, 27%, and 22%. The 1-year TRM was 16%, 18%, and 23%; the adjusted 3-year leukemia-free survival (LFS) was 54%, 48%, and 47%; and 3-year OS was 63%, 59%, and 54%. Multivariable analysis confirmed that consolidation was not prognostic for LFS. Similarly, consolidation was not associated with OS, relapse, TRM, or GVHD.

In a cooperative prospective study set up in seven countries, very-high-risk ALL in CR1 cases were defined by the presence of at least one of the following criteria: (1) failure to achieve CR after the first four drug induction phase; (2) t(9;22) or t(4;11) clonal abnormalities; and (3) poor response to prednisone associated with T immunophenotype, WBC of 100 × 10^9^/L or greater, or both. The primary outcome was disease-free survival, and analysis was by intention to treat. A total of 357 patients entered the study, of whom 280 were assigned chemotherapy and 77, related-donor HSCT. 5-year disease-free survival was 40.6% in those treated with chemotherapy and 56.7% in those assigned transplantation; 5-year survival was 50.1% and 56.4%. Although the combination of chemotherapy with TKI is considered the standard of care in patients with Ph-positive ALL, little is known about the impact of additional cytogenetic abnormalities (ACAs). Therefore, after retrospectively evaluating 1375 adult patients who underwent their first allogeneic HSCT in the TKI era, Akahoshi et al. reported that 224 patients had ACAs (16.3%). The ACAs that were seen in more than 20 cases (1.5%) were as follows: −7, del(22), del(9), +8, and +X. OS at 4 years was 56.9% in the group with ACAs and 60.5% in the group without ACAs. The cumulative incidence of relapse at 4 years was 28.9% in the group with ACAs and in the group with Ph alone. In multivariate analyses, there were no statistically significant differences in the risk of overall mortality or risk of relapse between the groups with and without ACAs.

A high-risk ALL patient profile is that with disease stage beyond the first CR, as well as that with CR1 and poor risk cytogenetics. This includes t(9;22), t(4;11), t(1;19), complex karyotype, and low hypodiploidy. Such a patient profile includes a slow response to induction chemotherapy, defined as more than 4 weeks to obtain CR, failure to achieve CR after induction, age over 35 or leukocytosis at diagnosis. Leukocytosis at diagnosis is defined as more than 30 × 10^9^/L in B-precursor ALL or more than 100 × 10^9^/L in T-cell ALL. Positive MRD is also associated with a poor outcome, even if the patient may or may not have morphological remission [90,91,92]. Autologous HSCT is not better than chemotherapy alone for the ALL patient because of its high relapse rate, according to the National Marrow Donor Program and Center for International Blood and Marrow Transplant Research of the National Institutes of Health in the US [93,94]; thus, an allogeneic HSCT is the best available therapy option for high-risk ALL patients, despite a high transplant-related mortality [95,96]. For ALL patients with a standard risk in the first CR, an HSCT has yet to prove to be superior to chemotherapy, but the UKALL/XII/Eastern Cooperative Oncology Group (ECOG) 2993 and the Dutch–Belgian Cooperative Trial Group for Hematology Oncology (HOVON)-18 ALL/HOVON-37 ALL studies have shown an OS for those that have received HSCT from a matched related donor [97]. Cumulative incidences of relapse at 5 years were, respectively, 24 and 55% for patients with a donor versus those without a donor. Non-relapse mortality (NRM) was estimated at 16% (+/−4) at 5 years after allogeneic HSCT. Thus, 5-year DFS was significantly better in the donor group: 60 versus 42% in the no-donor group. After risk-group analysis, improved outcome was more pronounced in standard-risk patients with a donor, with an OS of 69% at 5 years. Even if a similar outcome was reported for all leukemia patients when comparing a related with an unrelated HSCT [98,99,100], the ones with an unrelated donor seemed to have a poorer outcome, as shown by a higher TRM [67].

## 6. Immunotherapy for ALL

Ph-R/R B-cell ALL patients also benefit from blinatumomab, which is an FDA-approved bispecific murine antibody that simultaneously targets CD3 and CD19. The CR rate after blinatumomab is 67%, but the therapy is often associated with CRS (high fever, nausea, headaches), hepatic, and neurological side effects, among others [101,102,103,104].

T-cell ALL has an intrinsic resistance to chemotherapy, and thus a poor prognosis. Immunotherapy has the potential to improve the outcome of such patients. As CD38 is expressed in low levels in both normal lymphoid and myeloid cells, it may be an accurate target. Daratumumab is an immunoglobulin G1k monoclonal antibody that binds to CD38, already approved for the therapy of R/R multiple myeloma [105,106,107]. In Philadelphia, Teachey et al. tested daratumumab in a large panel of T-ALL patient-derived xenografts and found great efficacy in most cases [108]. In Jerusalem, Ganzel et al. further investigated the role of anti-CD38 immunotherapy [109], by treating a 14 year old boy diagnosed with R/R B-ALL. The response was good, with limited toxicity. However, after 6 months, he relapsed again and was administered bortezomib and gemtuzumab ozogamycin. In the post-transplant setting, daratumumab may play a role, as shown by Bonda et al. [110], who successfully saved a 32 year old young man with an early precursor T-ALL, relapsed after a previous allogeneic HSCT. Still, only isolated case reports have been published to date, and more extensive clinical trials are needed before assessing the role of daratumumab for ALL.

The anti-CD20 monoclonal antibody rituximab has had a substantial role in the improvement of outcome in Burkitt leukemia/lymphoma. In the US, Thomas et al., by treating 282 young adults with de novo Ph-negative precursor B-cell ALL with either standard or modified hyper-CVAD regimens, proved that the incorporation of rituximab in the hyper-CVAD regimen improves the outcome of CD20-postitive Ph-ALL [111]. The CR rate was 95%, with 3-year rates of CR duration (CRD) and OS of 60% and 50%, respectively. In the younger (age < 60 years) CD20-positive subset, rates of CRD and OS were superior with the modified hyper-CVAD and rituximab regimens compared with standard hyper-CVAD (70% vs. 38%). In contrast, rates of CRD and OS for CD20-negative counterparts treated with modified versus standard hyper-CVAD regimens were similar (72% vs. 68% and 64% vs. 65%). Older patients with CD20-positive ALL did not benefit from rituximab-based chemoimmunotherapy (rates of CRD 45% vs. 50% and OS 28% vs. 32%, respectively), related in part to deaths in CR. The data were confirmed in Europe by GRAAL investigators on 206 patients treated between 2006 and 2014 [112]. After a median follow-up of 30 months, EFS was longer in the rituximab group than in the control group (hazard ratio, 0.66; 95% CI, 0.45 to 0.98; *p* = 0.04); the estimated 2-year EFS rates were 65% and 52%, respectively. Treatment with rituximab remained associated with longer EFS in a multivariate analysis. Under investigation is now the use of inotuzumab ozogamycin (InO), a monoclonal antibody anti-CD22, which was studied in adults with R/R ALL [113,114]. In the MDACC study coordinated by Kantarjian et al., the median age was 36 years (range 6–80). CD22 was expressed in more than 50% of blasts in all patients. The median number of courses was two (range 1–5) and the median time between courses was 3 weeks (range 3–6). A total 18% of patients had CR, 39% had marrow complete response, 39% had resistant disease, and 4% died within 4 weeks of starting treatment. The OS was 57%. The most frequent adverse events during course one of treatment were fever (grade 1–2 in 20 patients, grade 3–4 in nine), hypotension (grade 1–2 in 12 patients, grade 3 in one), and liver-related toxic effects (bilirubin: grade 1–2 in 12 patients, grade 3 in two; raised aminotransferase concentration: grade 1–2 in 27 patients, grade 3 in one). Four years later, the same group reported that the rate of CR was significantly higher in the InO group than in the standard-therapy group (80.7% vs. 29.4%). Among the patients who had CR, a higher percentage in the InO group had results below the threshold for MRD (0.01% marrow blasts) (78.4% vs. 28.1%). In the survival analysis, which included all 326 patients, PFS was significantly longer in the InO group (median, 5.0 months vs. 1.8 months); the median OS was 7.7 months vs. 6.7 months, and the hazard ratio was 0.77. In the safety population, the most frequent grade 3 or higher non-hematological adverse events with InO were liver related. Veno-occlusive liver disease of any grade occurred in 15 patients (11%) who received InO and in 1% who received standard therapy. The study showed significantly higher rates of CR, PFS, and OS in the InO group vs. standard chemotherapy. Epratuzumab is another CD22 monoclonal antibody that has been studied in R/R ALL and investigated as part of a salvage therapy regimen both as a single-agent and in combination with standard reinduction chemotherapy [7]. 

For B-cell ALL, breakthroughs in immunotherapy have given new insights into the clinical management of such cases, reporting previously unprecedented CR rates. Immunotherapy is linked to the clinical development of anti-CD20 naked monoclonal antibodies rituximab, ofatumumab, and obinutuzumab; anti-CD19 ADCs SAR3419 and SGN-CD19A and anti-CD19 bispecific antibody blinatumomab; anti-CD22 naked monoclonal antibody epratuzumab and anti-CD22 ADC inotuzumab ozogamycin; anti-CD52 naked monoclonal antibody alemtuzumab; and anti-CD19 CAR-T cells [115]. By adding rituximab to B-ALL, the chemotherapy protocol has improved the outcomes of younger patients with CD20+ Ph-negative ALL [112]. Maury et al. report that, between 2006 and 2014, a total of 209 patients were enrolled in a large, multicentric study: 105 in the rituximab group and 104 in the control group. After a median follow-up of 30 months, EFS was longer in the rituximab group than in the control group; the estimated 2-year EFS rates were 65% and 52%, respectively. Treatment with rituximab remained associated with longer EFS in a multivariate analysis. The overall incidence rate of severe adverse events did not differ significantly between the two groups, but fewer allergic reactions to asparaginase were observed in the rituximab group. 

Chimeric antigen receptor-modified (CAR)-T cells are genetically engineered T cells that recognize unprocessed antigens. A B lymphocyte will recognize a native (unprocessed) antigen via surface immunoglobulins and then produce secreted antibodies. A T lymphocyte usually recognizes a processed antigen, usually a peptide, associated with MHC proteins on the surface of antigen-presenting cells [116,117,118,119]. The technology of producing CAR-T cells is continuously evolving, with fourth generation CAR-Ts constructed to include a cytokine expressing cassette [54,55,120,121]. For CAR-T-cell therapy, the T cells are collected, genetically engineered to express an artificial T cell receptor, and then infused back into the patient. Methods for gene delivery range from viral vectors to RNA-based methods. The use of viral vectors has the added benefit of resulting in permanent gene expression, thus perpetuating the antitumor activity throughout the lifetime of the CAR-T cells. For the moment, CD19 targeting CAR-T cells have been approved by FDA for R/R B-cell ALL (tisagenlecleucel) and R/R diffuse large B-cell lymphoma (axicabtagene ciloleucel). The success of CD-19 targeting CAR-T was made possible due to the specific and nearly universal expression of CD19 on B-lymphoblasts, making it an ideal target for immunotherapy [122,123]. Recently, CAR-T cell application has been expanded to CD22-positive B-cell ALL, with early preclinical studies showing antitumor activity in in vitro and in vivo models [124,125].

CAR-T cells are expected to bring the next big leap forward in leukemia immunotherapy [126], as two products have been approved in the US for B-cell malignancies. Tisangenlecleucel in a single perfusion has provided durable remission with long-term persistence in both pediatric and young adult patients with R/R B-cell ALL, with transient high-grade toxic effects [127]. In the global study, 75 patients received an infusion of tisagenlecleucel. 

Therapies based on CAR-T cells may have a huge potential clinical impact. Still, these new therapeutic alternatives may not be the “golden bullet,” as several mechanisms of resistance to therapy have been described. One such mechanism is the downregulation of antigen expression by leukemia cells [128,129]. Thus, Oak et al., from Stanford University Medical Center, presented their data on 22 patients treated with axicabtagene ciloleucel for B-cell lymphomas at the 2018 Annual Meeting of the American Society of Hematology in San Diego. At Day 28 post-administration of the autologous CAR product, the ORR was 86%: 10 patients had a complete response, 9 had a partial response, 1 had stable disease, and 2 had progressive disease. Both patients (2 of 2) with progressive disease at Day 28 had either dim or partial CD19 expression prior to CAR-T infusion, but nonetheless demonstrated robust Axi-cel expansion. One patient with Day 28 stable disease showed no CAR-T cell expansion, despite intact CD19 expression. Overall, there was no statistical difference in relative or absolute CAR-T cells in patients who responded versus those who did not at Day 28. At Day 90, out of the 22 patients, 5 patients (26%) developed progressive disease, and 4 of 5 underwent repeat biopsy. Of these patients, 2 had complete loss of tumor CD19 and another had downregulation of CD19 with variable expression of other B-cell antigens [130] CAR-T cell expansion was noted in multiple patients, suggesting that there may have been T-cell-intrinsic causes of treatment failure. Further studies are necessary to help identify and predict which patients will benefit from targeted immunotherapy [131,132].

## 7. Measurable Residual Disease

The optimal assessment of ALL therapy is measurable residual disease (MRD) measurement. The detection limit with currently available flow cytometry technology is around 1:100.000 malignant cells identified in mononucleated cells. The first attempts to rigorously measure MRD by flow cytometry were published in the late 80s and early 90s [133,134] on two laser cytometers, followed by polymerase chain reaction (PCR)-based MRD detection. Consequently, studies from the late 90s showed that monitoring MRD during the first months of therapy is a powerful indicator for outcome, regardless of whether it is done by PCR-based methods or by flow cytometry [135,136]. 

An important role in homogenizing and standardizing the flow-cytometry-based detection of MRD in acute leukemia was proposed by the European BIOMED-1 Concerted Action, which defined protocols for identifying normal subsets of B, T, and myeloid cells in bone marrow, the starting point for subsequent studies of the aberrant immunophenotypes involved in ALL. These efforts provided an excellent basis for standardized flow cytometric MRD studies in multicenter international treatment protocols for precursor-B-cell ALL and T-cell ALL patients, first by defining a normal pattern, and second, by establishing pathology-based patterns and optimal reagent combinations [137,138,139,140,141,142,143,144,145,146,147,148,149,150,151,152,153,154]. 

Current methods of flow cytometry MRD detection are still under improvement. It is under debate which independent markers or combinations of markers are more reliable. CD58 high expression is specific for B-leukemic blasts and thus, according to Shaver et al., it could lead to a cheaper and more time efficient approach [155]. 

Next generation sequencing (NGS)-based technology comprises all the developments of sequencing technologies, from Sanger sequencing to the actual methods that are used predominantly for research, but also in the clinical setting. The most important advantages of this technique are the possibility of identifying a larger spectrum of mutations and assessing different genomes without bias and higher sensitivity, thus allowing the identification of low frequency variants. Nevertheless, one of the major drawbacks of NGS is the large number of data that result from these kinds of experiment, which need storage and analysis by bioinformaticians [156,157,158].

In the Johns Hopkins study, Borowitz et al. used Ficoll–Hypaque-purified BM samples examined by RT-PCR. The presence of the common translocations *E2A-PBX1*, *TEL-AML1*, *BCR-ABL*, and *MLL-AF4* was evaluated. Other *MLL* gene rearrangements were detected by fluorescence in situ hybridization (FISH) studies using a break-apart probe strategy [159]. In Europe, Conter et al. have used screening by PCR amplification using the BIOMED-1 primer sets for Ig kappa deleting element gene rearrangements, *IGK*-Kde (Vk-Kde, intron-Kde), complete and incomplete TCR delta (*TCRD*; Vd-(Dd)-Jd1, Dd2-Jd1, Vd2-Dd3, Dd2-Dd3), and TCR gamma (*TCRG*; Vg-Jg1.3/2.3, Vg-Jg1.1/2.1) rearrangements [160]. 

Recently, NGS has been used for MRD evaluation in ALL, as it evaluated numerous V-(D)-J rearrangements linked to residual disease, but also to the normal immune repertoire of that specific cell [161,162,163,164]. Kotrova et al. used an NGS-based method to evaluate MDR in 76 patients with ALL and reported that by using NGS, they could predict more precisely the relapse at Day 33 of treatment than by PCR. By NGS, the clonal heterogeneity of the IgH locus has been evaluated, not possible by PCR or flow cytometry [165]. Furthermore, Pulsipher et al. compared NGS-based MRD with flow cytometry MRD in B-cell ALL patients that underwent HSCT. They showed that NGS-based MRD is superior to flow cytometry in predicting relapse and non-relapse in both pre- and post-HSCT samples [166]. Gawad et al. also used NGS-MRD evaluation of patients before and after HSCT transplant in blood samples, and confirmed its feasibility to evaluate relapse in patients where BM samples are not available [167]. 

Both flow cytometry and PCR are assays sensitive to MRD analysis, with deep clinical impacts [168,169]. Still, Denys et al., by analyzing 363 ALL patients at Days 15, 33 and 78, showed that 6-color flow cytometry significantly improves MRD analysis in ALL, but remains less sensitive than PCR-based MRD evaluation [170]. In clinical practice, this is important, especially following the approval of novel drugs, as is the case of blinatumomab, a drug that targets MRD-positive ALL. Gökbuget et al. showed that patients with MRD-positive B-cell precursor ALL had significantly longer RFS and OS when compared with MRD non-responders after treatment with blinatumomab [171]. 

Nowadays, MRD is the strongest independent prognostic factor in ALL. It is detected by molecular methods that use leukemia-specific or patient-specific molecular markers (fusion gene transcripts, or immunoglobulin/T-cell receptor (IG/TR) gene rearrangements), as well as by multi-parametric flow cytometry. Once MRD-based follow-up becomes more efficient [172,173,174,175], so does the effect of novel therapies, as is the case of blinatumomab or CAR-T cells, as presented in the article. 

## 8. Conclusions

The introduction of TKIs in the first-line chemotherapy for Ph-positive ALL has improved the clinical outcome for these patients. These patients were historically associated with cure rates of less than 25% in the pre-TKI era, but present-day long-term survival has reached more than 75% OS with the introduction of the novel approaches: immunotherapy, ponatinib, HSCT, and CAR-T cells. The best method for follow-up in ALL is MRD assessment. MRD is the measurement of the residual leukemia cells in patients after treatment. The threshold for MRD negativity is one malignant clone in 100,000 mononucleated cells, evaluated by either PCR or flow cytometry. MRD assessment is extensively employed in studies and clinical trials ranging from small molecules to CAR-T cells due to its significant prognostic and predictive value. 

Assessing the role of blinatumomab for the treatment of ALL and its potential to replace allogeneic SCT, moving from ‘bridge to transplant’ to ‘transplant replacement’ is the major “hot topic” in ALL research, expected to change the standard-of-care for this disease in the following years, with or without CAR-T cells.

## 9. Practice Points

ALL is classified as B-cell ALL and T-cell ALL, according to the WHO classification, and mixed-lineage ALL is a rare form of ALL with malignant cells displaying both B-cell and T-cell characteristic antigens [1,2]. The flow cytometry diagnosis of ALL is based on cells positive for CD10, CD19, CD20, CD22, CD24, and CD79a. The diagnostic workup is further supported by cytogenetics. Thus, hyperploidy or t(12;21)(p13;q22) is associated with a better prognosis, whereas t(9;22)(q34;q11.2), t(1;19)(q23;p13.3) or t(4;11)(q21;q23) translocations are associated with worse outcomes.

The backbone of chemotherapy regimens consists of systemic cytarabine, cyclophosphamide, methotrexate, and dexamethasone (among others), as well as CNS prophylaxis with intrathecal chemotherapy or irradiation. Tyrosine kinase inhibitors are effective in Ph-positive ALL and should be added to the chemotherapy protocol. Allo-HSCT is indicated upfront after the first remission in ALL patients with high-risk genetic abnormalities and in relapsed cases. Still, certain subgroups of relapsed or refractory ALL patients benefit from several recently introduced therapies, including blinatumomab (bispecific anti CD3-CD19), inotuzumab ozogamycin (anti CD22), and CD19-targeting CAR-T cells. Present-day disease evaluation and follow-up uses MRD assessment, which is increasingly used in the setting of clinical trials and is currently under translation in the routine clinical practice. The threshold for MRD is set to 1 malignant cell in 100,000 normal mononucleated cells, and it can be assessed with flow cytometry, PCR, or NGS.

## 10. Future Directions


Clarify the clinical significance of the provisional entities: B-cell ALL with intrachromosomal amplification of chromosome 21, and BCR-ABL1-like B-cell ALL.Clarify the appropriate therapy strategies for adolescents and young adult ALL patients, which fall between the standard categories of pediatric or adult ALL.Expand the use of CAR-T cells and assess the use of CAR-T cells as upfront therapy.Establish the role of MRD monitoring beyond studies and clinical trials.Set up the best standard-of-care for adolescents and young adults with ALL.Set up the best standard-of-care for older adults with ALL.Define the best treatment options or Ph-positive ALL in the era of TKIs.


## Figures and Tables

**Figure 1 jcm-08-01175-f001:**
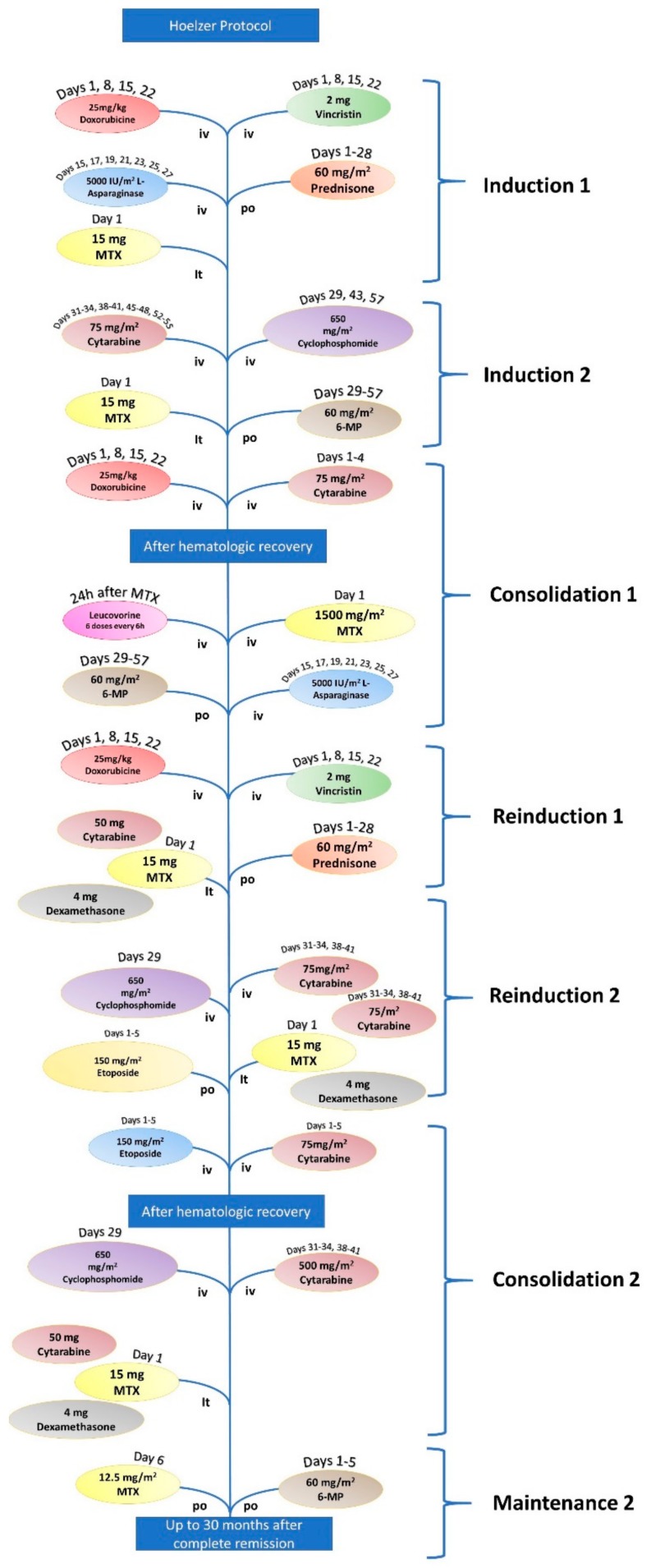
Workflow of the first-line therapy Hoelzer protocol for patients under the age of 65 years diagnosed with B-cell ALL (B-ALL).

**Figure 2 jcm-08-01175-f002:**
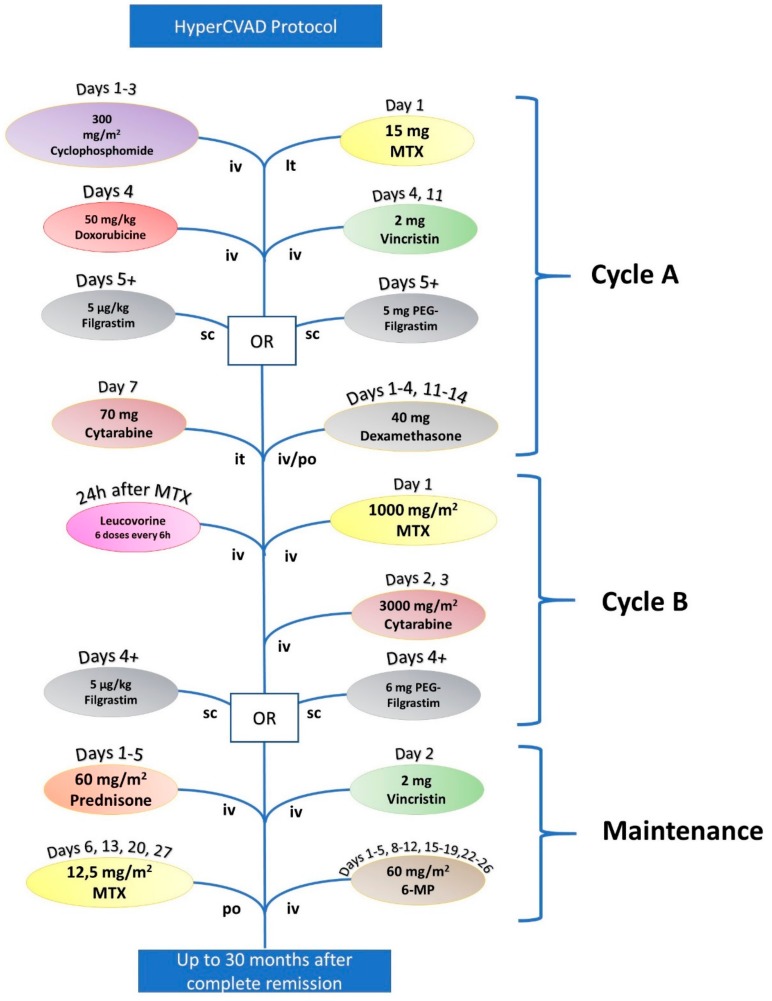
Workflow of the alternative treatment protocol, HyperCVAD, for patients under the age of 65 diagnosed with ALL.

**Figure 3 jcm-08-01175-f003:**
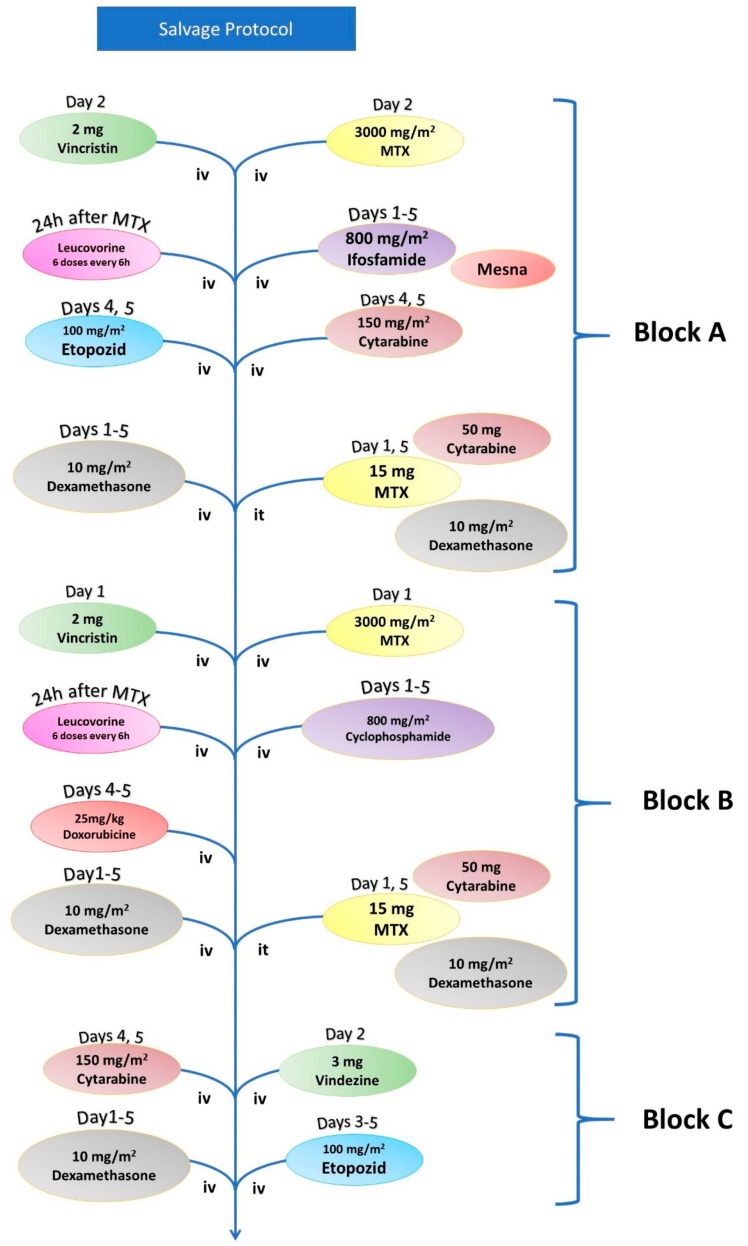
Workflow of the salvage chemotherapy regimen for patients under the age of 65 with relapsing ALL.

**Figure 4 jcm-08-01175-f004:**
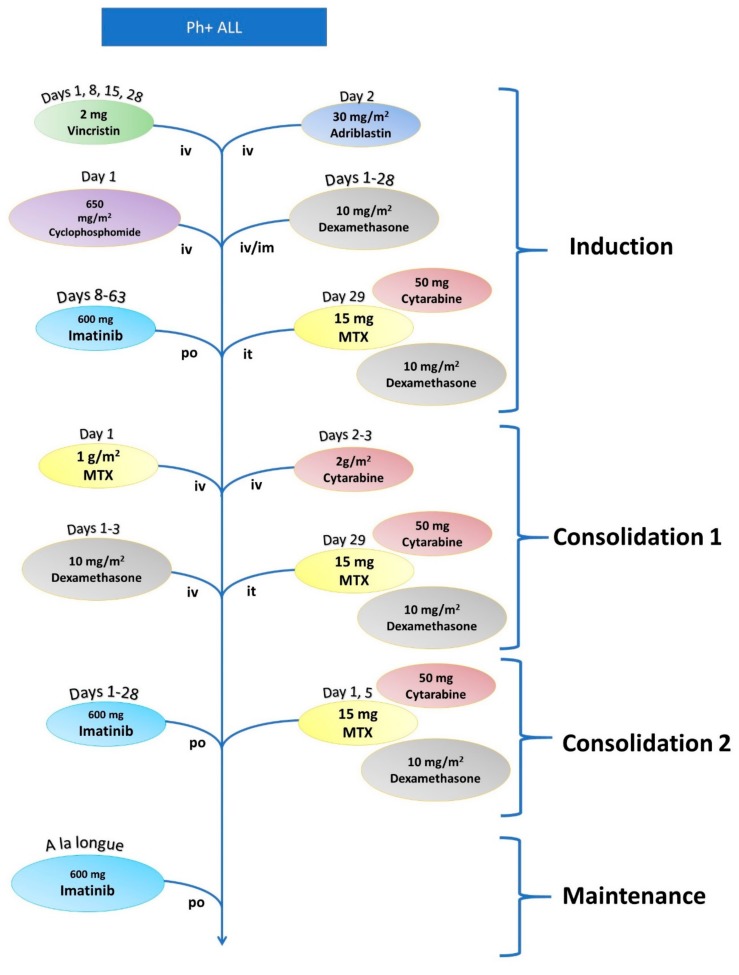
Workflow for the chemotherapy protocol for Philadelphia-positive ALL patients.

**Figure 5 jcm-08-01175-f005:**
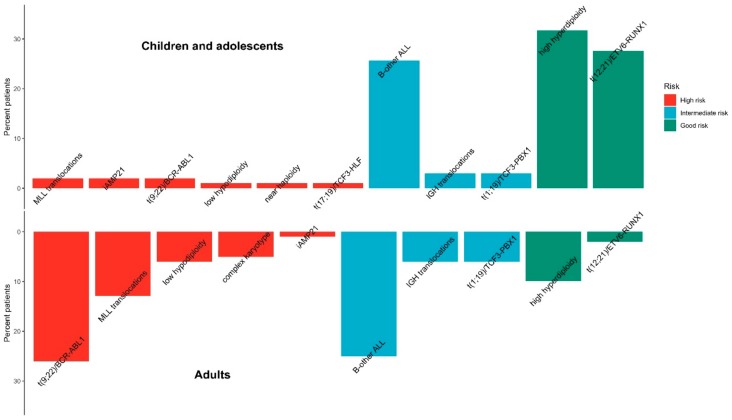
Percent distribution of ALL patients in terms of risk factor and cytogenetic profile.

**Table 1 jcm-08-01175-t001:** First-line therapy protocol for patients under 65 years diagnosed with acute lymphoblastic leukemia (ALL).

Hoeltzer Protocol for ALL			
Induction 1	Vincristine	2 mg	Days 1, 18,15, and 22
	Doxorubicin	25 mg/m^2^	Days 1, 8, 15, and 22
	L-Asparaginase	5.000 UI/m^2^	Days 15, 17, 19, 21, 23, 25, and 27
	Prednisone	60 mg/m^2^	Days 1–28
	Methotrexate (it)	15 mg	Day 1
Induction 2	Cyclophosphamide	650 mg/m^2^	Days 29, 43, and 57
	Cytarabine	75 mg/m^2^	Days 31–34, 38–41, 45–48, and 52–55
	6-mercaptopurine	60 mg/m^2^	Days 29–57
	Methotrexate (it)	15 mg	Days 31, 38, 45, and 52
Consolidation 1	Cytarabine	1000 mg/m^2^	Days 1–4
	Doxorubicin	30 mg/m^2^	Days 3–5
	Methotrexate + Leucovorin	1500 mg/m^2^	
	L-Asparaginase	10.000 UI/m^2^	Days 2 and 16
	6-mercaptopurine	25 mg/m^2^	Days 1–5 and 15–19
Reinduction 1	Vincristine	2 mg/m^2^	Days 1, 8, 15, and 22
	Doxorubicin	2 mg/m^2^	Days 1, 8, 15, and 22
	Prednisone	60 mg/m^2^	Days 1–28
	Methotrexate (it)	15 mg	Day 1
	Cytarabine (it)	50 mg	Day 1
	Dexamethasone (it)	4 mg	Day 1
Reinduction 2	Cyclophosphamide	650 mg/m^2^	Day 29
	Cytarabine	75 mg/m^2^	Days 31–34 and 38–41
	6-thiguanine	60 mg/m^2^	Days 29–57
	Methotrexate (it)	15 mg	Day 1
	Cytarabine (it)	50 mg	Day 1
	Dexamethasone (it)	4 mg	Day 1
Consolidation 2	Etoposide	100 mg/m^2^	Days 1–5
	Cytarabine	150 mg/m^2^	Days 1–5
Maintenance	6-mercaptopurine	60 mg/m^2^	Days 1–5
	Methotrexate	12.5 mg/m^2^	Days 1–5

**Table 2 jcm-08-01175-t002:** HyperCVAD protocol for ALL, an alternative to the Hoelzer protocol for patients under 65 years.

HyperCVAD			
Cycle A	Cyclophosphamide	300 mg/m^2^	Days 1, 2, and 3
	Doxorubicin	50 mg/m^2^	Day 4
	Vincristine	2 mg	Days 4 and 11
	Filgrastim or pegylated Filgrastim	5 μg/kg (Filgrastim) or 6 mg (PEG Filgrastim)	Starting Day 5
	Dexamethasone	40 mg	Days 1–4 and 11–14
	Cytarabine (it)	70 mg	Day 7
	Methotrexate (it)	15 mg	Day 2
Cycle B	Methotrexate (+Leucovorin)	1000 mg/m^2^	Day 1
	Cytarabine	6000 mg/m^2^	Days 2 and 3
	Filgrastim or PEG Filgrastim	5 μg/kg (Filgrastim) or 6 mg (PEG Filgrastim)	Starting Day 4 (Filgrastim) or at Day 4 (PEG Filgrastim)
Maintenance	Vincristine	2 mg	Day 2
	Prednisone	60 mg/m^2^	Days 1–5
	6-mercaptopurine	60 mg/m^2^	Days 1–5, 8–12, 15–19, and 22–26
	Methotrexate (it)	12.5 mg/m^2^	Days 6, 13, 20, and 27

**Table 3 jcm-08-01175-t003:** For relapsed ALL patients under 65 years of age, the salvage chemotherapy regimen is based on three blocks of administration with methotrexate.

Salvage Chemotherapy			
Block A	Vincristine	2 mg	Day 1
	Methotrexate (+Leucovorin)	3000 mg/m^2^	Day 1
	Ifosfamide (+Mesna)	800 mg/m^2^	Days 1–5
	Etoposide	100 mg/m^2^	Days 4 and 5
	Cytarabine	150 mg/m	Days 4 and 5
	Dexamethasone	10 mg/m^2^	Days 1–5
	Methotrexate (it)	15 mg	Day 1 and 5
	Cytarabine (it)	50 mg	Day 1 and 5
	Dexamethasone (it)	8 mg	Day 1 and 5
Block B	Vincristine	2 mg	Day 1
	Methotrexate (+Leucovorin)	3000 mg/m^2^	Day 1
	Cytarabine	200 mg/m	Days 1–5
	Doxorubicin	25 mg/m^2^	Days 4 and 5
	Dexamethasone	10 mg/m^2^	Days 1–5
	Methotrexate (it)	15 mg	Day 1 and 5
	Cytarabine (it)	50 mg	Day 1 and 5
	Dexamethasone (it)	8 mg	Day 1 and 5
Block C	Vindesine	3 mg	Day 1
	Cytarabine	2000 mg/m^2^	Day 1
	Etoposide	150 mg/m^2^	Days 3–5
	Dexamethasone	10 mg/m^2^	Day 1–5

**Table 4 jcm-08-01175-t004:** Chemotherapy protocol for Ph+ ALL.

Philadelphia Positive ALL			
Induction	Vincristine	2 mg	Days 1, 8, 15, and 28
	Adriblastin	30 mg/m^2^	Days 1–3
	Cytarabine	1200 mg/m^2^	Day 1
	Dexamethasone	8 mg/m^2^	Days 1–28
	Imatinib	600 mg	Days 8–63
	Methotrexate (it)	15 mg	Day 29
	Cytarabine (it)	50 mg	Day 29
	Dexamethasone (it)	8 mg	Day 29
Consolidation 1	Methotrexate	1 g/m^2^	Day 1
	Cytarabine	2 g/m^2^	Days 2–3
	Dexamethasone	8 mg/m^2^	Days 1–3
	Methotrexate (it)	15 mg	Day 29
	Cytarabine (it)	50 mg	Day 29
	Dexamethasone (it)	8 mg	Day 29
Consolidation 2	Imatinib	600 mg	Days 1–28
	Cytarabine (it)	50 mg	Day 1
	Dexamethasone (it)	8 mg	Day 1
Maintenance	Imatinib	600 mg	*A la longue*

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
