# Peer review of "Approach to the Adult Acute Lymphoblastic Leukemia Patient"

_jcm, 2019, doi:10.3390/jcm8081175_

Round 1

Reviewer 1 Report

Journal: Journal of Clinical Medicine
Manuscript ID: jcm-548004

The Review by Sas et. al.  contributes to an important knowledge domain with direct implications for patient care. The molecular basis, diagnosis, classification and treatment of ALL are reviewed. Recent publications and current protocols are considered. Although the significance of the work is high, the enthusiasm for this manuscript is diminished by its presentation.

Major points:

1.       The Title and the entire manuscript need to be revised for clarity.

2.       The abstract and the body of the review are not parallel. Revision of both is required.

3.       The molecular mechanisms responsible for ALL pathology need to be described in greater detail before the description of the chemotherapy regimens.

4.       Page 6, line 161, “intelligence” should be replaced with “cognitive functions”

5.       The conclusion “Thus, “old” or classic assays, based on cytology, should in 2019 be used mainly for screening, while more sensitive protocols as flow cytometry or genetic-based assays should decide whether an ALL patient should change therapy line or be addressed to the stem cell transplantation unit.” Is not supported by evidence. The claim should be substantiated. Additionally, “addressed” in “ALL patient should change therapy line or be addressed” should be changed to “referred”

6.       Research agenda section needs to be expanded.

Minor points:

1.       Page 1, line 6: “Cluj Napoca, Romania” among authors.

2.       All affiliations need to be proofread. For example: Is  “Carol Davila University of Medicine and Pharmacy” in Cluj Napoca ? -  page 1. Line 23

3.       Likewise,  1 is both “Department of Hematology, Iuliu Hatieganu University of Medicine and Pharmacy, Cluj-Napoca, Romania” and "Department of Experimental Therapeutics, National Cancer Center Hospital, Tokyo, Japan “  - lines 9 and 36

4.       Page 1, lines 36-47 need revision

5.       Page 7, starting with line 186 is italic. Why?

6.       Page 14 line 537: capitalize the first letter. Practice points section should be written with paragraphs or transformed into a table.

Author Response

Dear Editor,

              Thank you very much for reviewing our manuscript. We appreciate the tremendous effort and time the reviewers devoted to improving our manuscript. We sincerely feel that their thoughtful comments have further strengthened the manuscript. Specific responses to each comment are presented in the Responses to the Reviewers. In the revised manuscript, revisions to the manuscript are indicated in red font. We hope that our responses to the reviewers’ comments and the revisions made to the manuscript satisfy all questions and concerns. 

              With my best regards,

            Ciprian Tomuleasa, M.D.

             Department of Hematology,

             Iuliu Hatieganu University of Medicine and Pharmacy, Cluj Napoca, Romania.

Comments to the Reviewers

Reviewer 1:

The Review by Sas et. al.  contributes to an important knowledge domain with direct implications for patient care. The molecular basis, diagnosis, classification and treatment of ALL are reviewed. Recent publications and current protocols are considered. Although the significance of the work is high, the enthusiasm for this manuscript is diminished by its presentation.

Major points:

1.       The Title and the entire manuscript need to be revised for clarity.

Thank you very much for an important feed-back. We have revised the title of the manuscript, which now is “Clinical approach to the acute lymphoblastic leukemia patient “. The rest of the manuscript was revised. Professional English proofing added to the value of the revised version, which now we consider more suitable for publication.

2.       The abstract and the body of the review are not parallel. Revision of both is required.

Thank you very much for an important feed-back. We have revised both the abstract, as well as the rest of the manuscript in red. We now believe that the manuscript is suitable for publication.

3.       The molecular mechanisms responsible for ALL pathology need to be described in greater detail before the description of the chemotherapy regimens.

Thank you very much for an important feed-back. In the chapter of “Background on ALL workup and follow-up”, we have rephrased and supplemented the introduction in red in the revised manuscript that. Thus, the revised chapter is “Acute leukemias are classified into acute myeloid (AML) and acute lymphoblastic leukemias (ALL), each form having a characteristic immunophenotype. Based on the cytological aspect of the blasts, there are three main types of acute lymphoblastic leukemias: L1, with small cells that have a large nucleus; L2, with larger pleomorphic blasts; and L3, with a highly basophilic cytoplasm. The cytological classification is now mostly replaced by the World Health Organization (WHO) classification, which divides ALLs into B-cell ALL and T-cell ALL. B- cell ALL and acute lymphoblastic lymphomas are malignancies with B-cell lymphoblasts (1-3). When the primary disease is diagnosed in a lymph node, the correct name is "acute lymphoblastic lymphoma". In B-ALLs, which represents around 85% of all pediatric ALLs, the bone marrow aspirate displays at least 25% bone marrow lymphoblasts (1-4). Despite tremendous improvements in understanding the molecular mechanism behind B-cell ALL (B-ALL), the prognosis of these patients is rather poor, especially for old or frail patients that are unable to withstand aggressive chemotherapy or allogeneic hematopoietic stem cell transplantation. Moreover, in the case of adult patients, the disease is often already disseminated in extramedullary sites, especially in the central nervous system. Still, for young adults, the prognosis has been significantly improved, as proven by the Group of Research on Adult ALL (GRAALL) randomized controlled trial, which explored the role of hyperfractionated cyclophosphamide (hyper-C) dose intensification in newly-diagnosed Philadelphia (Ph)-negative ALL patients on a chemotherapy regimen similar to that of pediatric patients (5). The complete remission (CR) was 91.9 % and for a median follow-up of 5.2 years, the 5-year event-free survival (EFS) and overall survival (OS) was 52.2% and 58.5%, respectively. In adolescents and young adults, aged 15-20 years, the use of full pediatric protocols is supported by many comparative studies, with long term-survival of almost 70% (6). “. We have also added that “During normal B cell maturation, CD34 is first down regulated, together with TdT, followed by CD10 and CD38, while CD45 expression is up regulated, as well as CD21 and CD22 (11244048). Most aberrancies are related to the coexpression /over or under expression of CD10, TdT, CD38, CD34, CD20, and cross lineage myeloid expression, while aberrant T cell antigen expression is less frequent. For T-cell ALL, myeloid and B cell aberrant expression are also rare, and the detection of residual cells by flow cytometry following therapy is based on asynchronous expression of antigens in comparison with a normal maturation pattern, as further discussed in the manuscript. “.

4.       Page 6, line 161, “intelligence” should be replaced with “cognitive functions”

Thank you very much for an important feed-back. We have replaced the word “intelligence” with the word “cognitive functions”.

5.       The conclusion “Thus, “old” or classic assays, based on cytology, should in 2019 be used mainly for screening, while more sensitive protocols as flow cytometry or genetic-based assays should decide whether an ALL patient should change therapy line or be addressed to the stem cell transplantation unit.” Is not supported by evidence. The claim should be substantiated. Additionally, “addressed” in “ALL patient should change therapy line or be addressed” should be changed to “referred”.

Thank you very much for an important feed-back. We consider that the reviewer is correct is saying that the last paragraph is not supported by evidence and thus have decided to delete it. Thus, in the revised manuscript we remove a sentence that might be confusing for the readers.

6.       Research agenda section needs to be expanded.

Thank you very much for an important feed-back. We have expanded the Research agenda, which now is “Research agenda:

·       Clarify the clinical significance of the provisional entities: B cell-ALL with intrachromosomally amplification of chromosome 21, and BCR-ABL1–like B cell-ALL.

·       Clarify the appropriate therapy strategies for adolescents and young adult ALL patients, which fall between the standard categories of pediatric or adult ALL.

·       Expand the use of CAR-T cells and assessing the use of CAR-T cells as upfront therapy

·       Establish the role of MRD monitoring beyond studies and clinical trials.

·       Set up the best standard-of-care for adolescents and young adults with ALL.

·       Set up the best standard-of-care for older adults with ALL.

·       Define the best treatment options or Ph-positive ALL in the era of TKIs.

Minor points:

1.            Page 1, line 6: “Cluj Napoca, Romania” among authors.

Thank you very much for an important feed-back. We have corrected the manuscript, with the author list. The first submitted version had a proofing error. Also, the final author position is given to the working group between Romanian and Japanese physicians: Romanian-Japanese Working Group of the Romanian Society for Bone Marrow Transplantation.

All affiliations need to be proofread. For example: Is “Carol Davila University of Medicine and Pharmacy” in Cluj Napoca? -  page 1. Line 23.

Thank you very much for an important feed-back. We have corrected this mistake and have replaced the word “Cluj Napoca” with “Bucharest”.

3. Likewise, 1 is both “Department of Hematology, Iuliu Hatieganu University of Medicine and Pharmacy, Cluj-Napoca, Romania” and "Department of Experimental Therapeutics, National Cancer Center Hospital, Tokyo, Japan “- lines 9 and 36.

Thank you very much for an important feed-back. We have corrected this proofing errors in the revised version of the manuscript.

4.       Page 1, lines 36-47 need revision

Thank you very much for an important feed-back. We have corrected this proofing errors in the affiliation of the corresponding author. The revised version is now “

Ciprian Tomuleasa, M.D.

Department of Hematology, Iuliu Hatieganu University of Medicine and Pharmacy, 21 December No 73rd Boulevard, 400124, Cluj-Napoca, Romania.

5.       Page 7, starting with line 186 is italic. Why?

Thank you very much for an important feed-back. The paragraph was in Italic in the submitted version because of an error in proofing. We have corrected the revised manuscript.

6.       Page 14 line 537: capitalize the first letter. Practice points section should be written with paragraphs or transformed into a table.

Thank you very much for an important feed-back. In the revised manuscript, the chapter on Practice points is “ ALL is classified as B-cell ALL and T-cell ALL, according to the WHO classification and mixed lineage ALL is a rare form of ALL with malignant cells displaying both B-cell and T-cell characteristic antigens (1,2). The flow cytometry diagnosis of ALL is based on cells positive for CD10, CD19, CD20, CD22, CD24 and CD79a. The diagnostics workup is further supported by cytogenetics. Thus, hyperploidy or t(12;21)(p13;q22) is associated with a better prognosis, whereas t(9;22)(q34;q11.2), t(1;19)(q23;p13.3) or t(4;11)(q21;q23) translocations are associated with worse outcomes.

The backbone of chemotherapy regimens consists of systemic cytarabine, cyclophosphamide, methotrexate and dexamethasone (among others), as well as CNS prophylaxis with intrathecal chemotherapy or irradiation. Tyrosine kinase inhibitors are effective in Ph- positive ALL and should be added to the chemotherapy protocol. Allo-HSCT is indicated upfront after the first remission in ALL patients with high-risk genetic abnormalities and in relapsed cases. Still, certain subtypes of relapsed or refractory ALL patients benefit from several recently introduced therapies, including blinatumomab (bispecific anti CD3-CD19), inotuzumab ozogamycin (anti CD22) and CD19 targeting CAR-T cells. Present-day disease evaluation and follow-up uses MRD assessment, that is increasingly used in the setting of clinical trials and it is currently under translation in the routine clinical practice. The threshold for MRD is set to 1 malignant cell in 100.000 normal mononucleated cells and it can be assessed with flow cytometry, PCR or NGS.

Reviewer 2 Report

In their review article “Approach to the acute lymphoblastic leukemia patient now that’s a real challenge,” Sas et al describe the current standards of care for ALL treatment and the different ways of assessing minimal residual disease (MRD). This manuscript contained useful information and figures about standard of care treatment and valuable considerations into different ways to assess MRD in patients. However, the authors did not make any new conclusions or put their own perspective or opinion into this review other than citing and summarizing studies that have been done to date. While this can be useful, this decrease the impact on the field of this review article. There are some other major and minor issues that, if resolved, would enhance the clarity of the study.

Major issues:

1.     The focus of the article is on adult ALL, so it is confusing when childhood ALL is being discussed in some areas and not in others. It would be better if the whole review focused on adult ALL as this is a more challenging and less well defined area of practice.

2.     Section titles do not accurately describe what is being discussed in that section. For example, “Background on ALL work-up” section contains not only ALL work-up, but also treatment regimens and comparing clinical trials on different standards of care. It would be more accurate and easier to read to split up into multiple sections with titles that accurately describe what is being discussed within them. For example, background, work-up, treatment, HSCT transplant, immunotherapy, CNS involvement, associated toxicities, etc.

3.     There are many repetitive paragraphs and information is repeated throughout the article. The manuscript needs to be written in a more concise and straight-forward manner as it is very hard to follow, with poor organization as written.

4.     Many instances of incorrect grammar or language choices or spelling errors.

5.     There is no new perspective or conclusions from this review article that cannot be found elsewhere, it is simply a list of studies that have been done by other groups. It would elevate the article for the authors to provide their own insight on the topic and draw new conclusions from the work that they are summarizing.

6.     The title is not good—Acute Lymphoblastic Leukemias generally have a good prognosis, and are not as challenging to treat as other leukemia types. A title more accureately reflecting manuscript contents would be helpful.

7. Text in some of the figures is so small as to nearly be illegible. 

Minor issues:

1.     Abbreviations are often used and not defined the first time they are used. For example, RIC-HSCT, MUDs, NRM, GRAAAL, TRM, R/R, etc. MDR is used multiple times instead of MRD.

2.     Unclear at many times whether the authors are discussing T- or B-cell ALL. This needs to be clarified and split up better throughout the article.

3.     The authors bring up concerns and toxicities of using CNS prophylaxis, but it would add to the paper if they included data on the effects on patient outcomes of not using CNS prophylaxis to compare to outcomes on using it. (Line 163-168) Would also be helpful to split into a separate section on CNS involvement in ALL.

4.     Statement in line 170 that CNS relapses are often diagnosed in adults is incorrect. CNS involvement and relapse is not the norm. Would help to include CNS relapse rates in adults.

5.     Some sentences or parts of sentences are italicized but this does not add anything to the paper. If the authors are going to use italics it needs to be for things that are major conclusions or that they want to draw attention to from the readers, which is not how they are being used in this article.

6.     Line 202-206 are repetitive of treatment regimens from earlier in the article. This would be fixed by better organization and split into sections as suggested above.

7.     Hard to read text on bar graph in Figure 5. Should put titles horizontal at the end of the bars to make it easier to read.

8.     Not defined what major molecular response is clinically.

9.     Claim that B-ALL has a very poor prognosis (line 410) is not correct. Has one of the best prognosis of all cancer types. Maybe specifically in adults with relapsed ALL but needs to be clearly defined what is being referred to.

10.  Address issues with resistance to CAR-T therapy in discussion of CAR-T, including down-regulation of antigen expression by leukemia cells.

11.  MRD section should be before treatment section as the term MRD is used multiple times throughout the treatment section and guides treatment decisions but is not discussed until the end of the article.

12.  Majority of the conclusion discusses MRD when it was only one page of the entire review article. Needs to reflect what is discussed throughout the entire article. Does not include any conclusion on all of the different treatment approaches that were discussed in the body of the paper.

13.  Incomplete sentences at the end of the conclusion need to be fixed. (Lines 528-531)

14.  Not sure what the research agenda section is, as none of this was addressed in the paper. Maybe should be titled future directions? Could also be left out of the paper completely since it doesn’t really add anything to the review.

Author Response

Dear Editor,

              Thank you very much for reviewing our manuscript. We appreciate the tremendous effort and time the reviewers devoted to improving our manuscript. We sincerely feel that their thoughtful comments have further strengthened the manuscript. Specific responses to each comment are presented in the Responses to the Reviewers. In the revised manuscript, revisions to the manuscript are indicated in red font. We hope that our responses to the reviewers’ comments and the revisions made to the manuscript satisfy all questions and concerns. 

              With my best regards,

            Ciprian Tomuleasa, M.D.

             Department of Hematology,

             Iuliu Hatieganu University of Medicine and Pharmacy, Cluj Napoca, Romania.

Reviewer 2: 

In their review article “Approach to the acute lymphoblastic leukemia patient now that’s a real challenge,” Sas et al describe the current standards of care for ALL treatment and the different ways of assessing minimal residual disease (MRD). This manuscript contained useful information and figures about standard of care treatment and valuable considerations into different ways to assess MRD in patients. However, the authors did not make any new conclusions or put their own perspective or opinion into this review other than citing and summarizing studies that have been done to date. While this can be useful, this decrease the impact on the field of this review article. There are some other major and minor issues that, if resolved, would enhance the clarity of the study.

Thank you very much for an important feed-back. We have revised the manuscript, in accordance to the reviewer’s suggestions. It is true that our manuscript summarizes studies that have been published, as many other manuscripts. Still, following the chapter on Conclusions, we have added two more chapters: Practice points and Future directions. Thus, the present manuscript brings forward a very educational format, summarizing the main areas of interest in the field, a format that might potentially be appreciated by the practicing physician. We have also very carefully reviewed and corrected the issues stressed out by all reviewers and consider that the manuscript is now suitable for publication.

Major issues:

1.     The focus of the article is on adult ALL, so it is confusing when childhood ALL is being discussed in some areas and not in others. It would be better if the whole review focused on adult ALL as this is a more challenging and less well-defined area of practice.

Thank you very much for an important feed-back. In the chapter on Chemotherapy regimens, we have added the sentence “As this manuscript focuses mainly on adult patients, the presented protocols do not include pediatric ones.  “. We have deleted most of the information on childhood ALL, but we have kept the phrases that refer to the clinical management of the teenagers and young adults because very often physicians treating adult patients diagnosed with a hematological malignancy also treat young adults with ALL, older than 18. In ALL chemotherapy, young adults aged 18-30 have specific management protocols, based on pediatric regimens. Thus, we have added in the revised manuscript that “In the last 20 years the overall survival (OS) of children diagnosed and treated for ALL has improved dramatically, with the same results yet to be achieved in adult patients. Still, clinical improvement was reported in young adults (age <30), where pediatric protocols have proven to be equally if not more effective than the Hoelzer or HyperCVAD protocols. More specifically, the BFM (Berlin-Frankfurt-Munster) protocol has improved RFS (51 vs. 35%, p = 0.027), but not OS (43 vs. 33%, p = 0.2). “. We have also added the paragraph “An important variable in ALL outcome is age. While adolescents and young adults (from 15/18 to 35/40 years old) are treated with higher intensity and higher cumulative doses of drugs, elderly patients require a less aggressive protocol based on much lower doses of corticosteroids, vincristine and asparaginase, with the avoidance of anthracyclines and alkylating agents to reduce treatment related mortality (TRM).“.

2.     Section titles do not accurately describe what is being discussed in that section. For example, “Background on ALL work-up” section contains not only ALL work-up, but also treatment regimens and comparing clinical trials on different standards of care. It would be more accurate and easier to read to split up into multiple sections with titles that accurately describe what is being discussed within them. For example, background, work-up, treatment, HSCT transplant, immunotherapy, CNS involvement, associated toxicities, etc.

Thank you very much for an important feed-back. We have thoroughly reviewed the manuscript, have split the large chapter into smaller and better presented ones. Thus, the chapter on Chemotherapy regimens was split into the following chapters: Standard chemotherapy regimens, Tyrosine kinase inhibitors for Ph-positive ALL and Allogeneic stem cell transplantation and immunotherapy for ALL.

3.     There are many repetitive paragraphs and information is repeated throughout the article. The manuscript needs to be written in a more concise and straight-forward manner as it is very hard to follow, with poor organization as written.

Thank you very much for an important feed-back. We have reviewed the manuscript and have deleted the redundant information. The revised version of the manuscript is much more straight-forward and easier to follow, eligible for publication.

4.     Many instances of incorrect grammar or language choices or spelling errors.

Thank you very much for an important feed-back. Professional English proofing improved the revised manuscript, which we consider is now suitable for publication.

5.     There is no new perspective or conclusions from this review article that cannot be found elsewhere, it is simply a list of studies that have been done by other groups. It would elevate the article for the authors to provide their own insight on the topic and draw new conclusions from the work that they are summarizing.

Thank you very much for an important feed-back. We have revised the manuscript, in accordance to the reviewer’s suggestions. It is true that our manuscript summarizes studies that have been published, as many other manuscripts. Still, following the chapter on Conclusions, we have added two more chapters: Practice points and Research agenda. Thus, the present manuscript brings forward a very educational format, summarizing the main areas of interest in the field, a format that might potentially be appreciated by the practicing physician. We have also very carefully reviewed and corrected the issues stressed out by all reviewers and consider that the manuscript is now suitable for publication.

6.     The title is not good—Acute Lymphoblastic Leukemias generally have a good prognosis and are not as challenging to treat as other leukemia types. A title more accurately reflecting manuscript contents would be helpful.

Thank you very much for an important feed-back. We have revised the title of the manuscript, which now is “Clinical approach to the acute lymphoblastic leukemia patient “.

7. Text in some of the figures is so small as to nearly be illegible. 

Thank you very much for an important feed-back. We have initially submitted the manuscript with the figures as separate .TIFF images. In the separate images, the text is large and easily readable. We kindly ask the editor to accept to publish the revised manuscript with larger images.

Minor issues:

1.     Abbreviations are often used and not defined the first time they are used. For example, RIC-HSCT, MUDs, NRM, GRAAAL, TRM, R/R, etc. MDR is used multiple times instead of MRD.

Thank you very much for an important feed-back. We have defined the abbreviations in the manuscript, in red.

2. Unclear at many times whether the authors are discussing T- or B-cell ALL. This needs to be clarified and split up better throughout the article.

Thank you very much for an important feed-back. We have organized the revised manuscript better and consider that now is suitable for publication.

The authors bring up concerns and toxicities of using CNS prophylaxis, but it would add to the paper if they included data on the effects on patient outcomes of not using CNS prophylaxis to compare to outcomes on using it. (Line 163-168) Would also be helpful to split into a separate section on CNS involvement in ALL.

Thank you very much for an important feed-back. In the revised manuscript, we have added a new chapter, on Management of CNS involvement. We have also added in the revised manuscript that “When using very high-dose MTX, Nathan et al report that dose adjustments were needed in 74 of the 88-patient cohort (43). Out of the 74 patients with dose-adjustments, 9 cases had impaired MTX clearance or renal disfunction, 5 had hepatic toxicities, one patient had seizures and one pulmonary toxicity. None of the patients died because of the therapy. Still, this therapy is less toxic in comparison to cranial radiation plus intrathecal MTX. Such, the patients treated with very high dose MTX had a stable verbal IQ and increase in performance IQ related to expected practice events over time, in comparison with the group treated with radiotherapy plus intrathecal MTX. “.

4.     Statement in line 170 that CNS relapses are often diagnosed in adults is incorrect. CNS involvement and relapse is not the norm. Would help to include CNS relapse rates in adults.

Thank you very much for an important feed-back. We have corrected in the revised manuscript the inconsistency. Now the paragraph is “Recurrences within the CNS usually coincide with or predict systemic relapse in the marrow and blood. Even if adults achieve CR in most cases, relapses are diagnosed in 7% of cases, especially in the elderly and high-risk population. “.

5.     Some sentences or parts of sentences are italicized but this does not add anything to the paper. If the authors are going to use italics it needs to be for things that are major conclusions or that they want to draw attention to from the readers, which is not how they are being used in this article.

Thank you very much for an important feed-back. Some sentences were in the submitted manuscript in Italics because of a proofing error, when the initial manuscript was formatted. We have corrected this mistake in the revised manuscript.

6.     Line 202-206 are repetitive of treatment regimens from earlier in the article. This would be fixed by better organization and split into sections as suggested above.

Thank you very much for an important feed-back. We have re-organized the manuscript and have divided the previous very large chapter of Chemotherapy regimens into three smaller chapter. Thus, the manuscript is much ore fluent and better organized. Large paragraphs have also been moved in the revised manuscript, to better organize the information presented.

7.     Hard to read text on bar graph in Figure 5. Should put titles horizontal at the end of the bars to make it easier to read.

Thank you very much for an important feed-back. We have redesigned Figure 5. Thus, it much easier to read.

8.     Not defined what major molecular response is clinically.

Thank you very much for an important feed-back. We have added that “MMR is defined as BCR-ABL1/ABL ratio less than 0.1% in the BM. “.

9.     Claim that B-ALL has a very poor prognosis (line 410) is not correct. Has one of the best prognosis of all cancer types. Maybe specifically in adults with relapsed ALL but needs to be clearly defined what is being referred to.

Thank you very much for an important feed-back. The reviewer is correct. Thus, we have deleted this sentence and in the revised manuscript, the paragraph now starts with the sentence “For B-cell ALL, breakthroughs in immunotherapy shed new insights into the clinical management of such cases, with reporting previously unprecedented CR rates.  “.

10.  Address issues with resistance to CAR-T therapy in discussion of CAR-T, including down-regulation of antigen expression by leukemia cells.

Thank you very much for an important feed-back. We have added that “Therapies based on CAR T cells may have a huge potential clinical impact. Still, these new therapeutic alternatives may not be the “golden bullet” as several mechanisms of resistance to therapy have been described. One such mechanism is the downregulation of antigen expression by the leukemia cells (29391449, 30915277). Thus, Oak et al, from Stanford University Medical Center presented in the 2018 Annual Meeting of the American Society of Hematology in San Diego their data on 22 patients treated with axicabtagene ciloleucel for B-cell lymphomas. At day 28 post-administration of the autologous CAR product, the ORR was 86%: 10 patients had complete response, 9 had a partial response, 1 had stable disease, and 2 had progressive disease. Both patients (2 of 2) with progressive disease at day 28 had either dim or partial CD19 expression prior to CAR-T infusion, but nonetheless demonstrated robust Axi-cel expansion. One patient with day 28 stable disease showed no CAR-T cell expansion despite intact CD19 expression. Overall, there was no statistical difference in relative or absolute CAR+ T cells in patients who responded versus those who did not at Day 28. At day 90, out of the 22 patients, five patients (26%) developed progressive disease, and 4 of 5 underwent repeat biopsy. Of these patients, 2 had complete loss of tumor CD19 and another had downregulation of CD19 with variable expression of other B cell antigens. Lack of CAR-T cell expansion was noted in multiple patients, suggesting that there may be T cell intrinsic causes of treatment failure. Further studies are necessary to help identify and predict which patients will benefit from targeted immunotherapy. .

11.  MRD section should be before treatment section as the term MRD is used multiple times throughout the treatment section and guides treatment decisions but is not discussed until the end of the article.

Thank you very much for an important feed-back. We understand the point of view of the reviewer, but respectfully we don’t agree. We consider that the chapter of MRD should be placed after the chapter of therapy, as we evaluate MRD after at least one line of therapy. Still, should the editor decide to move the chapter of MRD before the chapters on therapy, we are more than happy to agree.

12.  Majority of the conclusion discusses MRD when it was only one page of the entire review article. Needs to reflect what is discussed throughout the entire article. Does not include any conclusion on all of the different treatment approaches that were discussed in the body of the paper.

Thank you very much for an important feed-back. In the revised manuscript, we have also discussed about bispecific antibodies and CAR T cells. Still, it is true that MRD is the main focus of the conclusion chapter as MRD is one of the main assays to follow-up ALL patients, helping physicians in their clinical decisions. One example of blinatumomab, a drug that is indicated for therapy for B-cell ALL with MRD-positive disease following at least one prior line of therapy.

13.  Incomplete sentences at the end of the conclusion need to be fixed. (Lines 528-531).

Thank you very much for an important feed-back. We have addressed this constructive critique in the revised manuscript. The conclusion is much better organized.

14.  Not sure what the research agenda section is, as none of this was addressed in the paper. Maybe should be titled future directions? Could also be left out of the paper completely since it doesn’t really add anything to the review.

Thank you very much for an important feed-back. We have changed the name of the chapter into Future directions. Still, we consider this chapter useful as it presents in a bullet point manner the areas of research and development for ALL, with potential clinical impact. Still, should the editor decide to delete them, we are more than happy to accept.

Reviewer 3 Report

Sas et al. aimed in this review to depict the approach to the ALL patient. It is usually expected from the authors of reviews that beyond writing summaries of scientific developments and ideas, they identify and discuss how the field may be impacted or develop in the future in an objective manner, including insights that may be of significance to the scientific community. In this reviewer opinion, the manuscript does not achieve this purpose.

Author Response

Dear Editor,

              Thank you very much for reviewing our manuscript. We appreciate the tremendous effort and time the reviewers devoted to improving our manuscript. We sincerely feel that their thoughtful comments have further strengthened the manuscript. Specific responses to each comment are presented in the Responses to the Reviewers. In the revised manuscript, revisions to the manuscript are indicated in red font. We hope that our responses to the reviewers’ comments and the revisions made to the manuscript satisfy all questions and concerns. 

              With my best regards,

            Ciprian Tomuleasa, M.D.

             Department of Hematology,

             Iuliu Hatieganu University of Medicine and Pharmacy, Cluj Napoca, Romania.

Reviewer 3: 

Authors of reviews that beyond writing summaries of scientific developments and ideas, they identify and discuss how the field may be impacted or develop in the future in an objective manner, including insights that may be of significance to the scientific community. In this reviewer opinion, the manuscript does not achieve this purpose.

Thank you very much for an important feed-back. We have revised the manuscript thoroughly, in accordance to all of the reviewer’s suggestions. It is true that our manuscript summarizes studies that have been published, as many other manuscripts. Still, following the chapter on Conclusions, we have added two more chapters: Practice points and Future directions. Thus, the present manuscript brings forward a very educational format, summarizing the main areas of interest in the field, a format that might potentially be appreciated by the practicing physician. We have also very carefully reviewed and corrected the issues stressed out by all reviewers and consider that the manuscript is now suitable for publication.

Round 2

Reviewer 1 Report

The authors adequately answered all my critique points.

Author Response

Dear Editor,

              Thank you very much for reviewing our manuscript. We appreciate the tremendous effort and time the reviewers devoted to improving our manuscript. We sincerely feel that their thoughtful comments have further strengthened the manuscript. Specific responses to each comment are presented in the Responses to the Reviewers. In the revised manuscript, revisions to the manuscript are indicated in red font. We hope that our responses to the reviewers’ comments and the revisions made to the manuscript satisfy all questions and concerns. 

              With my best regards,

            Ciprian Tomuleasa, M.D.

             Department of Hematology,

             Iuliu Hatieganu University of Medicine and Pharmacy, Cluj Napoca, Romania.

Comments to the Reviewers

Reviewer 1:

The authors adequately answered all my critique points. 

Thank you very much for an important feed-back. The revised manuscript is much better organized, and we consider that is now ready for publication.

Reviewer 2 Report

This revised manuscript is much improved! My only comment would be the split section “5. Allogeneic stem cell transplantation and immunotherapy for ALL.” into two separate sections. The review has a lot of useful information in it now and the authors have made all of the changes we suggested, as well as added better insight. 

Author Response

Dear Editor,

              Thank you very much for reviewing our manuscript. We appreciate the tremendous effort and time the reviewers devoted to improving our manuscript. We sincerely feel that their thoughtful comments have further strengthened the manuscript. Specific responses to each comment are presented in the Responses to the Reviewers. In the revised manuscript, revisions to the manuscript are indicated in red font. We hope that our responses to the reviewers’ comments and the revisions made to the manuscript satisfy all questions and concerns. 

              With my best regards,

            Ciprian Tomuleasa, M.D.

             Department of Hematology,

             Iuliu Hatieganu University of Medicine and Pharmacy, Cluj Napoca, Romania.

Comments to the Reviewers

Reviewer 2:

This revised manuscript is much improved! My only comment would be the split section “5. Allogeneic stem cell transplantation and immunotherapy for ALL.” into two separate sections. The review has a lot of useful information in it now and the authors have made all of the changes we suggested, as well as added better insight. 

Thank you very much for an important feed-back. The suggestion is very good. Thus, we have split the chapter into two separate chapters: “5. Allogeneic stem cell transplantation for ALL. “ and “6. Immunotherapy for ALL. “.

Reviewer 3 Report

The title should be changed to "Approach to the adult acute lymphoblastic leukemia patients"

Author Response

Dear Editor,

              Thank you very much for reviewing our manuscript. We appreciate the tremendous effort and time the reviewers devoted to improving our manuscript. We sincerely feel that their thoughtful comments have further strengthened the manuscript. Specific responses to each comment are presented in the Responses to the Reviewers. In the revised manuscript, revisions to the manuscript are indicated in red font. We hope that our responses to the reviewers’ comments and the revisions made to the manuscript satisfy all questions and concerns. 

              With my best regards,

            Ciprian Tomuleasa, M.D.

             Department of Hematology,

             Iuliu Hatieganu University of Medicine and Pharmacy, Cluj Napoca, Romania.

Comments to the Reviewers

Reviewer 3:

The title should be changed to "Approach to the adult acute lymphoblastic leukemia patients".

Thank you very much for an important feed-back. The suggestion is very good. The title was changed, according to the suggestions of the reviewer.
